# Awareness and Practices towards Vaccinating Their Children against COVID-19: A Cross-Sectional Study among Pakistani Parents

**DOI:** 10.3390/healthcare11172378

**Published:** 2023-08-23

**Authors:** Zain Ul Harmain, Noorah A. Alkubaisi, Muhammad Hasnain, Muhammad Salman, Mohamed A. Baraka, Zia Ul Mustafa, Yusra Habib Khan, Tauqeer Hussain Mallhi, Johanna C. Meyer, Brian Godman

**Affiliations:** 1Department of Medicines, Tehsil Head Quarter (THQ) Hospital, Fortabbas 62020, Pakistan; zain.ul.harmain@gmail.com; 2Department of Botany and Microbiology, College of Science, King Saud University, Riyadh 11451, Saudi Arabia; nalkubaisi@ksu.edu.sa; 3Department of Medicine, Tehsil Head Quarter (THQ) Hospital, Kallurkot, Bhakkar 30000, Pakistan; hasnainazads@gmail.com; 4Institute of Pharmacy, Faculty of Pharmaceutical and Allied Health Sciences, Lahore College for Women University, Lahore 54000, Pakistan; msk5012@gmail.com; 5Department of Pharmacy, Fatima College of Health Sciences, Abu Dhabi 64141, United Arab Emirates; 6Clinical Pharmacy Department, College of Pharmacy, Al-Azhar University, Cairo 11651, Egypt; 7Discipline of Clinical Pharmacy, School of Pharmaceutical Sciences, Universiti Sains Malaysia, Penang 11800, Malaysia; 8Department of Pharmacy Services, District Headquarter (DHQ) Hospital, Pakpattan 57400, Pakistan; 9Department of Clinical Pharmacy, College of Pharmacy, Jouf University, Sakaka 72388, Saudi Arabia; yhkhan@ju.edu.sa (Y.H.K.); thhussain@ju.edu.sa (T.H.M.); 10Department of Public Health Pharmacy and Management, School of Pharmacy, Sefako Makgatho Health Sciences University, Ga-Rankuwa 0208, South Africa; hannelie.meyer@smu.ac.za (J.C.M.); or brian.godman@smu.ac.za (B.G.); 11South African Vaccination and Immunisation Centre, Sefako Makgatho Health Sciences University, Ga-Rankuwa 0208, South Africa; 12Department of Pharmacoepidemiology, Strathclyde Institute of Pharmacy and Biomedical Science (SIPBS), University of Strathclyde, Glasgow G4 0RE, UK

**Keywords:** COVID-19, vaccines, vaccine hesitancy, parents, children, acceptance, Pakistan

## Abstract

There are typically lower COVID-19 vaccination rates among developing versus higher-income countries, which is exacerbated by greater vaccine hesitancy. However, despite the increasing evidence of safety, parents are still reluctant to vaccinate their children against COVID-19. This is a concern in countries experiencing successive waves, such as Pakistan. Consequently, the objective of this study was to gain better understanding and practice regarding parents vaccinating their children against COVID-19 in Pakistan. A cross-sectional study was conducted to measure parents’ attitudes towards vaccinating their children. In total, 451 parents participated in the study, giving a response rate of 70.4%; 67.4% were female, 43.2% belonged to the 40–49 years age group, and 47.7% had three children, with 73% of parents fully immunized against COVID-19. We found that 84.7% of parents did not consider COVID-19 to be a very serious issue, and 53.9% considered that their children were not at high risk of COVID-19. Overall, only a quarter of the study participants had currently vaccinated their children and 11.8% were willing to vaccinate their children in the near future. Parents who had a better knowledge of COVID-19, secondary or higher education, children who had chronic illness, and those parents whose children had been infected with COVID-19 were more likely to have their children vaccinated. The most common reasons for vaccine hesitancy were “my child is not at high risk of COVID-19” (61%) and “I am afraid to put/inject a foreign object inside my child’s body” (52.2%). Overall, vaccine acceptance was low among the parents of the children. Those parents with higher education, chronic illnesses, greater knowledge of COVID-19 and its vaccines, and those whose children had been infected with COVID-19 were significantly (*p* < 0.001) inclined towards vaccinating their children. Effective campaigns as well as awareness sessions are needed to address misinformation and reduce vaccine hesitancy.

## 1. Introduction

Since the emergence of the SARS-CoV-2 virus and the associated Coronavirus disease of 2019 (COVID-19) in 2019 in Wuhan, China, an increasing number of cases have been reported throughout the world in all age groups [1,2,3]. More than 765 million positive cases and 6.9 million deaths globally have been documented up to 8 May 2023 [4]. According to the American Academy of Pediatrics, almost 15.6 million positive cases of COVID-19 have been reported in children, along with more than 8000 cases being reported per day up to 4 May 2023 [5]. The overall prevalence of COVID-19 among children has been 17.9% since the pandemic began [5]. A multi-national study revealed a 10% hospital mortality rate in children critically ill with COVID-19 admitted to hospital, and this was higher in those with cardiac and pulmonary co-morbidities [6]. Whilst mortality rates are lower in hospitals in low- and middle-income countries (LMICs) when considering all the children admitted with COVID-19 [7,8,9], systematic reviews have suggested that mortality rates are generally higher among children hospitalized with COVID-19 and subsequently admitted to intensive care units in LMICs [10].

In Pakistan, the first positive case of COVID-19 was reported on 26 February 2020. Following this, a significant number of positive cases have been seen in different waves among all age groups in the country [11,12]. Overall, more than 1.5 million positive cases of COVID-19 and 30,634 deaths have been reported in Pakistan up to 10 May 2023 [13].

Unfortunately, there is currently no central data base or registry to document COVID-19 cases or deaths among children aged between 1 and 18 years in Pakistan. However, local media reported that 19,367 children between the ages of 1 and 18 had tested positive with COVID-19 between March 2020 and March 2021 in Lahore, the capital city of Punjab Province [14]. Similarly, 5792 children up to the age of 10 years tested positive for COVID-19 from April 2020 to April 2021 in the capital city, Islamabad [15]. A previous study from Punjab Province indicated that among neonates and children admitted with COVID-19 in different referral hospitals, 3.2% of admitted neonates subsequently died [16]. This is a concern that needs to be addressed among this population.

The lack of effective treatments at the start of the pandemic for patients with COVID-19, despite many medicines including hydroxychloroquine, lopinavir-ritonavir, ivermectin and remdesivir being proposed and used, along with the significant economic impact of lockdown measures and the health consequences of the pandemic, resulted in considerable activities among scientists to develop safe and effective vaccines against COVID-19 [17,18,19,20,21,22,23,24,25]. There has also been considerable over-prescribing of antibiotics in patients admitted to hospital with COVID-19 despite limited evidence of bacterial co-infection and secondary infection [26,27,28]. This includes patients admitted to hospital in Pakistan with COVID-19, which includes neonates and children [12,29,30]. Such practices are an additional concern as they increase antimicrobial resistance (AMR), further enhancing morbidity, mortality, and costs [31,32,33,34,35,36].

The government of Pakistan initiated the first formal COVID-19 vaccination campaign on 2 February 2021, and in the first phase of the vaccination drive, front-line healthcare workers (HCWs) were prioritized to receive COVID-19 vaccines free of charge [37]. Following this, multiple vaccine centers were established in every district, and in the next phase older individuals over 60 years received COVID-19 vaccines free-of-charge at these vaccine centers [38]. In the subsequent phase, citizens over 50 years were vaccinated against COVID-19 and the rigorous vaccination campaign continued across the country whereby any citizen above 18 years could be vaccinated in a nearby vaccination center. The fact that the COVID-19 vaccines were free-of-charge to citizens in Pakistan was important to enhance vaccination rates. Otherwise, the costs of the vaccine would be prohibitive for an appreciable number of citizens and thus severely detrimental to the success of any campaign [39]. In September 2022, the vaccination of children above five years was initiated throughout the country and parents were advised to vaccinate their children [40,41]. Parents could subsequently vaccinate their children at established vaccine centers in every district and tehsil level. Moreover, vaccination camps were established at schools and public places, along with mobile vaccine vehicles, to vaccinate the majority of the children above the age of five years. As a result, this demonstrated the commitment of the Government of Pakistan to vaccinating this age group. To facilitate the vaccination program, Pakistan had a wide range of COVID-19 vaccines approved by the health authorities available for public use. The vaccines included the BBIBP-CorV vaccine (Sinopharm), CoronaVac vaccine (Sinovac), Ad5-nCoV vaccine (CanSino), mRNA-1273 vaccine (Moderna), ChAdOx1-S vaccine (AstraZeneca), BNT162b2 vaccine (Pfizer-BioNTech), Gam-COVID-Vac vaccine (Sputnik V), and the Pakistani-made Pakvac vaccines [42].

Currently, nearly half of the population of Pakistan is below the age of 18 years. Consequently, in order to vaccinate the majority of the country’s population, this age group must be considered as a priority [43]. However, hesitancy towards available vaccines is a major barrier to complete immunization in LMICs [44]. Parental vaccine hesitancy is defined as “delay in accepting or refusing vaccines for their children despite their availability” [45]. Hesitancy towards COVID-19 vaccines have been reported globally, particularly among LMICs, enhanced by misleading and false information spread through social media platforms [46,47,48,49], with vaccine hesitancy in parents hindering vaccination programs among children [45]. This is a concern as low vaccine uptake among children affects their health as well as that of the community [45]. The discredited study of Wakefield, which was published in 1998, regarding an alleged link between the MMR vaccine and autism still has an impact today [50]. Recent publications have documented continued declines in MMR vaccine coverage rates, resulting in increased rates of measles across countries [51,52]. This situation needs avoiding for COVID-19 vaccines among children, as vaccination rates influenced by parents.

We are aware from previous studies undertaken in Pakistan that COVID-19 vaccine hesitancy has been reported among the general population [53], health care workers [54], patients hospitalized with COVID-19 [55], pregnant women [56], and patients on maintained hemodialysis [57]. However, to the best of our knowledge, no study has been undertaken to ascertain the awareness and practices of COVID-19 vaccines among parents for their children below the age of 18 years. Consequently, we sought to address this evidence gap and its implications by conducting a multicenter cross-sectional study to document parents’ attitudes towards COVID-19 vaccination for their children. Subsequently, use the findings from this study to provide future guidance to all key stakeholder groups in Pakistan and beyond.

## 2. Materials and Methods

### 2.1. Study Design and Setting and Procedures

A cross-sectional study design was employed, with data gathered from the parents of children aged 5–18 years using a validated study instrument. This age group was chosen because the Government of Pakistan in 2022 initiated a vaccination campaign in this age group [40,41]. Children below the age of 5 years were not included in the current vaccination campaign. The current study was conducted in four districts in Punjab Province, namely Sahiwal, Pakpattan, Faisalabad, and Vehari. This was out of 36 total districts over three months (May–July 2022). Punjab Province was chosen for this initial study as it represents the majority of country’s population [12,30]. As mentioned, multiple COVID-19 vaccines centers were established in every district in Punjab Province where all citizens above age 5 years can obtain COVID-19 vaccines free-of-charge.

### 2.2. Study Instrument

Data were collected using a previously validated study instrument in Iraq and Jordan following permission from the corresponding authors. Before using the instrument, slight modifications were made in line with previous studies conducted in Pakistan by the co-authors and others [55,56,57,58]. A pilot study was conducted among randomly selected 20 parents in order to confirm the clarity and understandability of the content of the questionnaire before utilization. All the participants of the pilot study confirmed the clarity of the study tool along with suggestions for slight modification. After incorporating the suggestions of participants from the pilot study, the final study questionnaire (Appendix A) consisted of following seven sections:

**Section-I** consisted of eleven questions including general information about the parents and their children. The requested demographics of participating parents included their age, gender, level of education, monthly income, and COVID-19 vaccination status. Among those who had not yet been vaccinated, their willingness to subsequently get vaccinated including their children.

**Section-II** dealt with the health status of parents and their children based on three questions. The three questions included the presence of any chronic diseases including diabetes mellitus, hypertension, and cardiovascular disease among parents. In the case of children this included those who had, or currently have, any chronic diseases including asthma, allergies, diabetes, cancer, cardiovascular disease, sickle cell anemia, liver disease, and thalassemia. In addition, this also included taking steroids or immunosuppressants for their condition.

**Section-III** consisted of three questions about parents’ experience with COVID-19. Parents were asked if they, their relatives, or their children had been infected (or not) with COVID-19 since the beginning of this breakout.

**Section-IV** had four questions related to the impressions of study participants about COVID-19. This included the potential seriousness of COVID-19 either in them or their children, and the likelihood that they think they would get COVID-19 in next 6 months.

**Section-V** contained 23 questions about their knowledge of COVID-19. This included the symptoms, preventive measures, ways of transmission, and practices adopted by parents to protect their children from getting COVID-19.

**Section-VI** had four questions about their knowledge of COVID-19 vaccines. This included knowledge regarding their safety, side effects and methods of administration.

**Section-VII** contained eleven questions about the willingness to vaccinate their children and the reasons for any vaccine hesitancy

Knowledge scores were collated and subsequently divided into poor (below a total of 50% of correct answers), moderate (50 to 75% correct answers), and good (over 75% correct answers).

Suggestions for future strategies to reduce hesitancy and increase vaccination among children in Pakistan were based on the considerable experiences of the co-authors working in this area across LMICs.

### 2.3. Sample Size Calculation

The sample size was calculated using Raosoft online sample size calculator assuming a 50% response rate from the study population, 95% confidence interval and 5% margin of error. The calculated sample size was 377.

### 2.4. Data Collection Procedure

The team of investigators approached parents of children at different places including COVID-19 vaccines centers, schools, and even public places such as markets and parks. The investigators subsequently invited them to participate in the survey. All the participants were briefed about the purpose and execution of the study before their participation in the study. Parents who were willing were supplied with study instruments and requested to provide their response. In case of any confusion, investigators facilitated the study participants in completing the survey. The participants were conveniently approached and their participation in the survey was entirely voluntary.

### 2.5. Statistical Analysis

Parents’ responses were entered in SPSS version 22 after appropriate coding. Categorical variables were presented as numbers and percentages. The normality of continuous data was assessed using the Shapiro–Wilk test. In addition, a histogram of continuous data was also examined, which showed a non-normal distribution of continuous variables. Consequently, median and interquartile ranges were used to present continuous data. The Chi-Square test was used to compare categorical data between the vaccine acceptors and rejecters. The Mann–Whitney U test was used to compare continuous data (COVID-19 knowledge and preventive practices score) between parents who were willing to vaccinate their children and those who were unwilling. A *p* value of less than 0.05 was taken as statistically significant.

### 2.6. Ethical Approval

Approval for the current study was obtained from the Office of Research, Innovation and Commercialization (ORIC), Lahore College for Women University, Jail Road, Lahore. Approval was also obtained from the COVID-19 vaccination centers/administrators before data collection. All participants provided written informed consent prior to their participation in the study. Participation was voluntary and all the participants were assured of the confidentiality of their data. Moreover, no personal data, including the names or identity numbers, of the study participants were collected by the investigators.

## 3. Results

The data of 451 parents (out of 640 approached) having a child between 5 to 18 years of age were included in the analysis. Demographic data of the study participants are given in Table 1. The majority of respondents were mothers (67.4%) and those belonging to the 40–49 years age group (43.2%). Regarding the education status, 7.1% were illiterate, 4% possessed religious education, 33.3% primary, 43.2% secondary, and 12.4% higher secondary or above level of education. Most of the study participants had a monthly household income of PKR 31,000–60,000 (USD 1 = PKR 285). The majority of parents sampled had three children (47.7%). 8.6% reported having a child who had a chronic illness. 72.5% of participating parents were fully immunized against COVID-19, whereas 14.9% were partially vaccinated, and 12.6% had not been vaccinated. Of these 57 unvaccinated parents, 18 showed willingness to receive the COVID-19 vaccine.

The data showed that 23.3% of participating parents reported contracting COVID-19 during the pandemic while 2.2% reported that their child had suffered from COVID-19. Overall, 84.7% of parents did not consider COVID-19 to be a very serious issue and 53.9% considered their children were not at high risk of COVID-19 (Table 2).

Participants’ responses to items assessing their knowledge of COVID-19 and its vaccines are given in Table 3. The median knowledge score was 13 (11, 15), with the majority of parents having a moderate level of COVID-19 related knowledge (50–75% knowledge scores = 68.1%). Approximately 17% of participating parents had poor knowledge of COVID-19 and its vaccines. Regarding preventive measures for COVID-19 practiced by the study parents during COVID-19 pandemic (Table 4), everyone reported using face masks when in public, 87.4% reported frequently washing their hands with soap, 86% used detergents/disinfectant/sanitizers most of the time, 92.9% frequently practiced social distancing, and 93.3% avoided touching their eyes/mouth/nose/face without cleaning their hands. The median COVID-19-preventive-practice score was 15, with almost all having an average level of COVID-19-related practices (99.8% having scores 50–75%).

As shown in Figure 1a, 25% (n = 113) of participating parents reported that they had vaccinated their 5-to 18-year-old children for COVID-19. Of the 338 parents with children that had not been immunized against COVID-19, only 11.8% were willing to vaccinate their children (Figure 1b). The most common reasons for COVID-19 vaccine refusal were “my child is not at high risk of COVID-19” (61%), “I am afraid to put/inject a foreign object inside my child’s body” (52.2%), and “this vaccine has not been adequately tested on children” (48.1%) (Figure 2).

Table 5 shows the differences in the socio-demographic characteristics between parents who were willing to vaccinate their children for COVID-19 and those who were not. Parents who had secondary or higher level of education were more inclined to vaccinate their children for COVID-19. In addition, parents whose child had a chronic illness and those whose children suffered from COVID-19 showed greater willingness to vaccinate their children. Those who had a higher level of knowledge regarding COVID-19 and its vaccines were also more inclined to accept COVID-19 vaccination for their children.

## 4. Discussion

We believe the present study is the first study to evaluate Pakistani parent’s knowledge, attitude, and practices towards COVID-19 vaccination among children during the current mass vaccination drive in the country including children aged 5 to 18 years [40,41], as well as the possible reasons for refusing COVID-19 vaccines for their children. Alongside this, potential socio-demographic characteristics of parents that influence their acceptance or refusal towards COVID-19 vaccination in their children.

We are aware that mortality from COVID-19 is currently lower in adults and children than other populations [59]. However, to date, over 17,000 children worldwide have died from COVID-19 [60]. In addition, in the USA, COVID-19 has been the seventh leading cause of death among children aged 5–11 years, and was an even higher cause for children aged 12–18 years, leading to calls to vaccinate all children above the age of 5 [61]. Consequently, encouraging vaccination in this population will help protect them and the community in the future [45]. This is especially important in LMICs where there are concerns with over-crowding, sub-optimal sanitation, and high levels of co-payment needed to see a physician and purchase medicines. These factors can enhance the presence of respiratory diseases alongside considerable purchasing of antibiotics without a prescription, especially ‘Watch’ antibiotics, thus exacerbating AMR [62,63,64,65]. AMR is a growing concern in Pakistan with ongoing national and regional efforts to reduce AMR through the National Action Plan; however, progress is currently hampered by personnel and resource issues [66].

Encouragingly, our study revealed that the majority of the parents possessed moderate knowledge of COVID-19 and its vaccines. These findings are in line with previous studies which reported about the general population, university students, medical students and health care providers in Pakistan, who typically possessed adequate awareness about COVID-19 [67,68,69,70]. Encouragingly as well, most of the study participants were practicing COVID-19 preventive measures including the use of face, hand hygiene, and social distancing. This is similar to another study from Pakistan which showed that the majority of the general population in Pakistan were practicing appropriate preventive measures against COVID-19 [71].

We believe the possible reasons for moderate, as opposed to poor, knowledge and practices regarding COVID-19 among participating parents is that government of Pakistan has instigated a number of measures to educate the public about COVID-19 since the start of the pandemic in the absence of effective treatments. These include broadcasting information regarding the transmission of COVID-19 from one person to another, the possible signs and symptoms of COVID-19 and effective preventive measures. Educational activities included awareness campaigns involving print and electronic media, announcements at public, religious and educational places, and personal messages sent to people’s mobile phones [72]. Moreover, the current study was conducted in the late phase (2022) of the pandemic, which will have enhanced the knowledge base of participating parents.

The parents’ impression towards COVID-19 in their children was surprising as most of them did not consider COVID-19 to be a very serious issue and more than half considered their children were not at high risk of COVID-19. This is in contrast to the findings of a recent study in Egypt where nearly half of parents surveyed considered COVID-19 a very serious threat to their child [73].

We are not sure of the reasons behind these differences and will be exploring this further in future. This is because the majority of the parents in our study were aware of the usefulness of vaccinations against COVID-19 as they were fully or partially vaccinated. There were similar findings in a multinational study conducted among eight eastern Mediterranean countries where the majority of the parents were vaccinated against COVID-19 [74].

However, there were concerns among the parents in our study around vaccinating their children against COVID-19 with only 25.1% of children currently vaccinated and only 11% willing to subsequently vaccinate their children. This could be due to a lack of enough evidence supporting the safety and efficacy of available COVID-19 vaccines in children, which is similar to studies in China [75] and Thailand [76]. This is despite evidence showing COVID-19 vaccines to be safe and effective at preventing symptomatic COVID-19 in children [77,78]. Alongside this, parents reported on a perception that their children were not at risk of getting COVID-19 infection, fear of injecting foreign objects into their children, and thinking that COVID-19 vaccines may cause autism in children. Since children are at high risk of getting infections including COVID-19, health authorities should convey the possible threats to the parents about risks of COVID-19 to themselves, their children, and their community going forward. In addition, healthcare authorities should comprehensively convey to parents the current efficacy and safety of COVID-19 vaccines among children as more data become available. Alongside this, parents should be provided with sufficient information about the increased risks of getting COVID-19 without comprehensive vaccination programs, and the implications for them, their children, and the community.

Interestingly, our findings regarding the extent of vaccine hesitancy among parents for their children in Pakistan are in direct contrast to studies reported from India wherein 85% of parents reported acceptance of COVID-19 vaccines for their children [79]. In addition, it was reported that in Malaysia 73.6% of parents were willing to have their children vaccinated [80], and in Turkey 36.3% of parents were willing to have their children vaccinated, rising to 83.9% if mortality rates are higher with new variants [81]. The situation in these LMICs provides hope for the future in Pakistan as more data become available.

We are aware of the limitations of our study. Firstly, we only collected data from Punjab Province for the reasons stated. Secondly, we only collected data from four districts of Punjab by using convenient sampling. Consequently, it is likely our findings will have some degree of bias. Thirdly, since the study was anonymous and we did not collect participants’ identification, we were unable to collect information about those who refused to join our study. Those who refused to participate in the study might have different characteristics compared to the study participants. Fourthly, health authorities are stressing parents to vaccinate their children, particularly school-going children and also requesting school administrations to lobby parents. Consequently, parents could be more inclined towards getting vaccinated their children against COVID-19 versus previous intentions. However, despite these shortcomings, we believe our findings are robust and will be useful in devising policies and strategies to maximize vaccination campaigns, particularly among children against COVID-19 in Pakistan and beyond.

## 5. Conclusions and Recommendations

Our study concluded that most of the parents of children aged 5–18 years possessed moderate knowledge regarding COVID-19 and vaccinations, as well as an average level of preventive practices concerning COVID-19. However, only a quarter of children in our study were immunized against COVID-19. In addition, only a limited number of the parents whose children have currently not been vaccinated were willing to have their children subsequently vaccinated. The most common factors affecting COVID-19 vaccine hesitancy among parents in Pakistan were false believes that their child was not at high risk of COVID-19, fear of injecting a foreign object inside their child’s body, as well as concerns with the safety and effectiveness of current vaccines due to inadequate testing on children. The health authorities in Pakistan must initiate appropriate educational campaigns through the engagement of different stakeholders to enhance COVID-19 vaccine uptake among children in the future, and this includes extensive social media activities. We will continue to monitor the situation.

## Figures and Tables

**Figure 1 healthcare-11-02378-f001:**
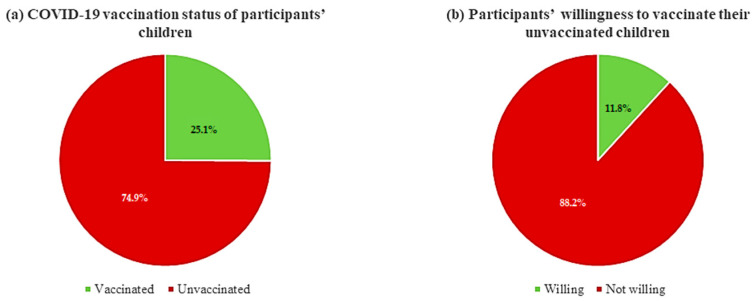
(**a**) COVID-19 vaccination status of participants’ 5-to 18-year-old children, (**b**) participants’ willingness to vaccinate their unvaccinated 5-to-18-year children for COVID-19.

**Figure 2 healthcare-11-02378-f002:**
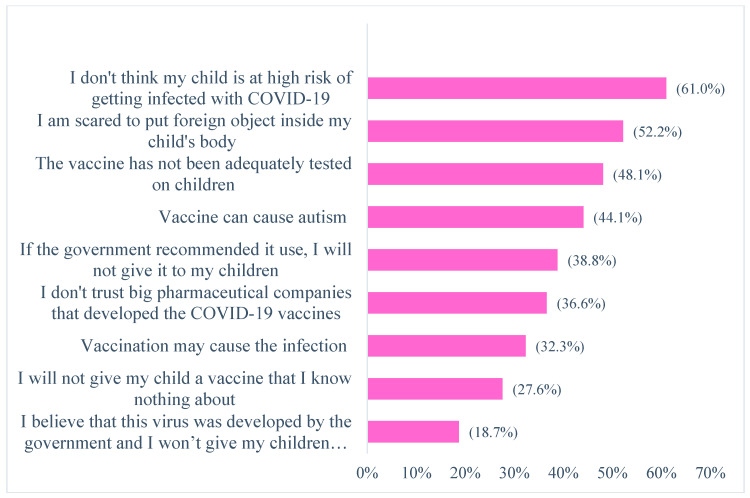
Reasons for refusing to vaccinate children against COVID-19.

**Table 1 healthcare-11-02378-t001:** Demographic data of parents having children between 5-to 18-years of age.

Variable	Subgroups	N (%)
**Age (years)**	<40	123 (27.3)
	40–49	195 (43.2)
	≥50	133 (29.5)
**Sex**	Male	147 (32.6)
	Female	304 (67.4)
**Education**	Illiterate	32 (7.1)
	Religious education only	18 (4.0)
	Primary	150 (33.3)
	Secondary	195 (43.2)
	Higher secondary or above	56 (12.4)
**Income (PKR)**	<30,000	29 (6.4)
	31,000–60,000	311 (69.0)
	>60,000	111 (24.6)
**Smoking status**	Smoker	51 (11.3)
	Non-smoker	400 (88.7)
**Working or studying in a medical field**	Yes	21 (4.7)
	No	430 (95.3)
**Any chronic disease**	Yes	67 (14.9)
	No	384 (85.1)
**COVID-19 Immunization status**	Fully immunized	327 (72.5)
	Partially immunized	67 (14.9)
	Unvaccinated	57 (12.6)
**Willing to take COVID-19 vaccine** **(Currently unvaccinated—N = 57)**	Yes	18 (31.6)
	No	27 (47.4)
	Maybe	12 (21.1)
**No. of children**	1	23 (5.1)
	2	166 (36.8)
	3	215 (47.7)
	4 or more	47 (10.4)
**Do you have any preterm children?**	Yes	50 (11.1)
	No	401 (88.9)
**Do any of your children suffer from a chronic illness or taking steroids or immunosuppressant medications?**	Yes	39 (8.6)
	No	412 (91.4)
**Do you know somebody close to you that was infected with COVID-19?**	Yes	451 (100.0)
	No	--
**Have you been infected with COVID-19 during the pandemic?**	Yes	105 (23.3)
	No	334 (74.1)
	Maybe	12 (2.7)
**Has any of your children ever been infected with COVID-19?**	Yes	10 (2.2)
	No	422 (93.6)
	Maybe	19 (4.2)

**Table 2 healthcare-11-02378-t002:** Participants’ impression towards COVID-19.

Variable	Sub-Groups	N (%)
**Estimated seriousness of COVID-19 on participants**	Low risk	382 (84.7)
	High risk	69 (15.3)
**What is the likelihood that you will be infected with COVID-19 during the next 6 months?**	I think that I will be infected and my symptoms will be severe	32 (7.1)
	I think that I will be infected and my symptoms will be mild	67 (14.9)
	I do not think that I will be infected	93 (20.6)
	I do not know	259 (57.4)
**Estimated seriousness of COVID-19 on children**	Low risk	243 (53.9)
	High risk	208 (46.1)
**What is the likelihood that your children will be infected with COVID-19 during the next 6 months?**	I think my child will be infected and my symptoms will be severe	191 (42.4)
	I think my child will be infected and my symptoms will be mild	66 (14.6)
	I do not think my child will be infected	89 (19.7)
	I do not know	105 (23.3)

**Table 3 healthcare-11-02378-t003:** Participants’ knowledge related to COVID-19 and its vaccine.

Items	N (%)	
	Correct	Incorrect
**Signs and symptoms**		
Fever	325 (72.1)	126 (27.9)
Chills	211 (46.8)	240 (53.2)
Cough	147 (32.6)	304 (67.4)
Diarrhea	308 (68.3)	143 (31.7)
Middle ear infection	221 (49.0)	230 (51.0)
Loss of smell and taste	278 (61.6)	173 (38.4)
No symptoms	428 (94.9)	23 (5.1)
**Transmission**		
Drinking unclean water	270 (59.9)	181 (40.1)
Eating unclean food	277 (61.4)	174 (38.6)
Inhalation of respiratory droplets	231 (51.2)	220 (48.8)
Eating or touching wild animals	175 (38.8)	276 (61.2)
**Preventive measures**		
Washing hands with regular soap	401 (88.9)	50 (11.1)
Using detergents	262 (58.1)	189 (41.9)
Social distancing	372 (82.5)	79 (17.5)
Avoid touching face/mouth/nose/eyes	266 (59.0)	185 (41.0)
Avoid eating meat	355 (78.7)	96 (21.3)
Consuming herbs	371 (82.3)	80 (17.7)
**Is there a drug in pharmacies or medical stores that can cure COVID-19?**	190 (42.1)	261 (57.9)
**Vaccine-related knowledge**		
Effectiveness of COVID-19 vaccine in children	155 (34.4)	296 (65.6)
Safety of COVID-19 vaccine in children	196 (43.5)	255 (56.5)
COVID-19 vaccine administration	378 (83.8)	73 (16.2)

**Table 4 healthcare-11-02378-t004:** Preventive practices related to COVID-19.

Preventive Practices	N (%)				
	Almost Always	Most of the Time	Sometimes	Rarely	Never
Wearing face masks when in public	--	451 (100.0)	--	--	--
Washing hands with soap	--	394 (87.4)	43 (9.5)	13 (2.9)	1 (0.2)
Using detergents	--	388 (86.0)	54 (12.0)	9 (2.0)	--
Social distancing	--	419 (92.9)	29 (6.4)	3 (0.7)	--
Avoid touching face/mouth/nose/eyes with contaminated/unclean hands	--	421 (93.3)	25 (5.5)	5 (1.1)	--

**Table 5 healthcare-11-02378-t005:** Comparisons of socio-demographic data between vaccine intending and not-intending parents.

Variable	Subgroups	Intention to Vaccinate 5–18 Years Old Children for COVID-19		*p*-Value
		Willing	Not Willing	
**Age (years)**	<40	10 (25.0)	81 (27.2)	0.930
	40–49	17 (42.5)	128 (43.0)	
	≥50	13 (32.5)	89 (29.9)	
**Gender**	Male	27 (67.5)	193 (64.8)	0.860
	Female	13 (32.5)	105 (35.2)	
**Education**	No formal education	0 (0.0)	46 (15.4)	**<0.001**
	Primary	5 (12.5)	131 (44.0)	
	Secondary or above	35 (87.5)	121 (40.6))	
**Income**	<30,000 PKR	3 (7.5)	19 (6.4)	0.076
	31,000–60,000 PKR	23 (57.5)	220 (73.8)	
	>60,000 PKR	14 (35.0)	59 (19.8)	
**Smoking status**	Smoker	7 (17.5)	30 (10.1)	0.175
	Non smoker	33 (82.5)	268 (89.9)	
**Working or studying in a medical field**	Yes	0 (0.0)	1 (0.3)	1.000
	No	40 (100.0)	297 (99.7)	
**COVID-19 Immunization status**	Fully immunized	30 (75.0)	217 (72.8)	0.051
	Partially immunized	9 (22.5)	39 (13.1)	
	Unvaccinated	1 (2.5)	42 (14.1)	
**Any chronic disease**	Yes	6 (15.0)	44 (14.8)	1.000
	No	34 (85.0)	254 (85.2)	
**Do you have any preterm children?**	Yes	2 (5.0)	31 (10.4)	0.399
	No	38 (95.0)	267 (89.6)	
**Do any of your children suffer from a chronic illness or taking steroids or immunosuppressant’s medications?**	Yes	7 (17.5)	21 (7.0)	**0.034**
	No	33 (82.5)	277 (93.0)	
**Have you ever been infected with COVID-19?**	Yes (confirmed/suspected)	2 (5.0)	31 (10.4)	0.399
	No	38 (95.0)	267 (89.6)	
**Has any of your children ever been infected with COVID-19?**	Yes (confirmed/suspected)	7 (17.5)	21 (7.0)	**0.034**
	No	33 (82.5)	277 (93.0)	
**Estimated seriousness of COVID-19 on participants**	Low risk	34 (85.0)	266 (89.3)	0.425
	High risk	6 (15.0)	32 (10.7)	
**What is the likelihood that you will be infected with COVID-19 during the next 6 months?**	I think that I will be infected	13 (32.5)	66 (22.1)	0.275
	I do not think that I will be infected	9 (22.5)	62 (20.8)	
	I do not know	18 (45.0)	170 (57.0)	
**Estimated seriousness of COVID-19 on children**	Low risk	19 (47.5)	162 (54.4)	0.500
	High risk	21 (52.5)	136 (45.6)	
**What is the likelihood that your children will be infected with COVID-19 during the next 6 months?**	I think my child will get infected	24 (60.0)	168 (56.4)	0.356
	I do not think my child will be infected	5 (12.5)	65 (21.8)	
	I do not know	11 (27.5)	65 (21.8)	
**Knowledge of COVID-19 and its vaccines**	--	212.66	163.71	**0.003**
**Preventive practice related to COVID-19**	--	163.44	170.31	0.614

## Data Availability

Additional data are available from the corresponding authors on reasonable request.

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
