# Peer review of "Awareness and Practices towards Vaccinating Their Children against COVID-19: A Cross-Sectional Study among Pakistani Parents"

_healthcare, 2023, doi:10.3390/healthcare11172378_

Round 1

Reviewer 1 Report

- Table 5, column Sig must be P-value.

- Figure 3, percent must be only in x-axis in parenthesis: percent (%)

- Figures 1 and 2, they should be smaler and with better resolution.

Author Response

Reviewer 1

Open Review

Quality of English Language

(x) I am not qualified to assess the quality of English in this paper
( ) English very difficult to understand/incomprehensible
( ) Extensive editing of English language required
( ) Moderate editing of English language required
( ) Minor editing of English language required
( ) English language fine. No issues detected

Yes

Can be improved

Must be improved

Not applicable

Does the introduction provide sufficient background and include all relevant references?

(x)

( )

( )

( )

Are all the cited references relevant to the research?

(x)

( )

( )

( )

Is the research design appropriate?

(x)

( )

( )

( )

Are the methods adequately described?

(x)

( )

( )

( )

Are the results clearly presented?

(x)

( )

( )

( )

Are the conclusions supported by the results?

(x)

( )

( )

( )

Comments and Suggestions for Authors

1) - Table 5, column Sig must be P-value.

Author comments: Thank you. We have updated it now.

2) - Figure 3, percent must be only in x-axis in parenthesis: percent (%)

Author comments: Thank you. We have updated it now.

3) Figures 1 and 2, they should be smaller and with better resolution

Author comments: Thank you. We have updated it now.

Reviewer 2 Report

I would than theauthors for this interesting work whichould help in the understandng of the phenomenon of vaccine hesitancy.

After reviewing I found some concerns which are shown in the comment below:

1. Even the introdction is too long, It has not allowed to define the question of the work. hat is the need to conduct this work? what are the results in the world?

2. The results needs to be reorganized and the determination of he risk factors should be supprtedby the resuts of the regression analysis (for example). (See below).

3. The discussion is too long and it contains multiple self-statements sentences. It contains also multiple repetetive and sometimes contradictory sentences.

4. THe manuscript should be edited forEnglish languae and proofead for spellig errors.

Below is all the details :

a. Add "Pakistan" to the title

b. Line 34: understanding of what? complete please

c. Cross-sectional questionaire (correct please; we can not say this, cros-sectional survey, quesionnaire based....).

c. Line 50: COVID-19 (correct in line 127)

d. Line 53: Coronavirus 

e. Line 69: untill 10

g. Line 77: delete "admitted were"

h. Line 78: correct: despite many....(it is not correct)

i. Line 89-90: delete " Free...citizens"

j. Line 113: to our knowledge, we do not believe (uncorrect)

k. Line 134: reorder the references according to their appearence in the text (Iraqi 39, Jordan 40...) (and in the list of references)

l. Line 142: and COVID-19

m. Line 193: 5-17 or 5-18 years?

n. Line 196-197: correct the sentence (have religious....)

o. table 1: you should reorder the factors of te table:

You should begin with the sociodemographics: (age, gender (use sexe please), education,..children, ...).

Then you return to: health status, COVID-19 infections, COVID-19 vaccination...).......

p. line 211: Yo should fine what do you consider as moderate knoweldge, poor, adequate, (in the conclusion you used sufficient, appropriate.....). Define these terms. what is the rate that alowed you to define the level as adequate, poor....?

Later how did you define the actors associated with the high level of knowledge?

In the same line: what does this expression mean ? "knowledge (50-75% knowledge scores = 211 68.1%)".

Figure 1, 2 and 3: delete the title frm the backgound of the figures.

Figure 1 and 2: reduce the size of the figures (you can make them in one line).

Figure 3: delete the vertical bars

improve is quality by making each item in 1 line only (exp the four last items). complete the item  and delete the points.

Line 246: corect please

Line 247: towards vaccination...

Lines 248-152: the sentences are incomplete

Line 256: practice preventive measures is not correct

Line 260-264: this may be also to the fact that the study was conducted in 2022 (not in 2020), effect od scial medias, medias....

Line 270: delete "is"

Line 271: delete the sentence "we are....studies".

Line273: delete "is'", correct "control". also you should delete al the paragraph of lines 273-76. (no relation)

Lines 278-281: there interminable number of studies about vaccine hesitancy....

Line 282: 83: correct the numbers (25.06....)

Line 282-294: multiple redundant and contradictory sentences. revise 

Line 307-312: delete the paragraph

Line 331:  "only a quarter of children in Pakistan" . i may be a quarter of the study population but not all the children of Pakistan.

In conclusion:

The manuscript requires an extensive revision

Extensive editing of English language required

Author Response

Reviewer 2

Open Review

Quality of English Language

( ) I am not qualified to assess the quality of English in this paper
( ) English very difficult to understand/incomprehensible
(x) Extensive editing of English language required
( ) Moderate editing of English language required
( ) Minor editing of English language required
( ) English language fine. No issues detected

Author comments: Thank you for this. The manuscript has been edited by one of the co-authors who is a native English speaker with over 500 publications in recent years in peer-reviewed Journals. We trust this is now acceptable.

Yes

Can be improved

Must be improved

Not applicable

Does the introduction provide sufficient background and include all relevant references?

( )

(x)

( )

( )

Are all the cited references relevant to the research?

( )

(x)

( )

( )

Is the research design appropriate?

( )

(x)

( )

( )

Are the methods adequately described?

( )

(x)

( )

( )

Are the results clearly presented?

( )

(x)

( )

( )

Are the conclusions supported by the results?

( )

(x)

( )

( )

Comments and Suggestions for Authors

I would thank the authors for this interesting work which could help in the understanding of the phenomenon of vaccine hesitancy.

Author comments: Thank you for these kind words – appreciated.

After reviewing I found some concerns which are shown in the comment below

Author comments: Thank you for your help. We hope we have adequately addressed the comments raised.

  1. Even the introdction is too long, It has not allowed to define the question of the work. What is the need to conduct this work? what are the results in the world?

Author comments: Thank you for this. We have now more clearly laid out the rationale for our study. This starts with providing general information on prevalence/ mortality rates with COVID-19 including children before moving onto Pakistan specifically. Subsequently – why vaccination against COVID-19 is so important, e.g. in the absence of effective treatments certainly initially despite all the hype – examples include HCQ, remdesivir and ivermectin – as well as overuse of antibiotics in patients with COVID-19 driving up AMR and its consequences – so need to avoid this. Vaccination in Pakistan includes children (after initial high priority groups) – however aware of possible hesitancy. Want to avoid e.g. another Wakefield situation – hence the need for more information in this group in Pakistan to help formulate future policies. We trust this is now acceptable. 

  1. The results needs to be reorganized and the determination of the risk factors should be supported by the results of the regression analysis (for example). (See below).

Author comments: Thank you. Table 5 showed socio-demographic factors between parents who were willing to vaccinate their children for COVID-19 and those who were not using Mann-Whitney U and Kruskal-Wallis H test.

  1. The discussion is too long and it contains multiple self-statements sentences. It contains also multiple repetitive and sometimes contradictory sentences.

Author comments: Thank you for this. We have now re-organised the Discussion to improve the flow, and trust this is now acceptable.

  1. THe manuscript should be edited for English language and proof read for spelling errors.

Author comments: Thank you for this comment. One of the co-authors who is a native English speaker with over 500 publications in peer-reviewed Journals has been through the manuscript and updater it. We trust it is now OK.

  1. Add "Pakistan" to the title

Author comments: Thank you. We have updated it now.

  1. Line 34: understanding of what? complete please

Author comments: Thank you. We have updated the manuscript and trust this is now acceptable.

  1. Cross-sectional questionaire (correct please; we can not say this, cros-sectional survey, quesionnaire based....).

Author comments: Thank you. We have updated it now.

  1. Line 50: COVID-19 (correct in line 127)

Author comments: Thank you. We have updated it now.

  1. Line 53: Coronavirus

Author comments: Thank you. We have updated it now.

  1. Line 69: untill 10

Author comments: Thank you. We have updated it now.

  1. Line 77: delete "admitted were"

Author comments: Thank you. We have updated it now.

  1. Line 78: correct: despite many....(it is not correct)

Author comments: Thank you. We have updated it now.

  1. Line 89-90: delete " Free...citizens"

Author comments: Thank you for this. We believe though this is very important for countries such as Pakistan where wages are low and without free vaccines – a vast majority of the population would not be vaccinated. We trust this is now OK.

  1. Line 113: to our knowledge, we do not believe (uncorrect).

Author comments: Thank you. We have updated it now.

  1. Line 134: reorder the references according to their appearence in the text (Iraqi 39, Jordan 40...)

(and in the list of references).

Author comments: Thank you. The references have now been updated.

  1. Line 142: and COVID-19

Author comments: Thank you. We have updated it now.

  1. Line 193: 5-17 or 5-18 years?

Author comments: Thank you. We have updated it now.

  1. Line 196-197: correct the sentence (have religious....)

Author comments: Thank you. We have updated it now.

  1. table 1: you should reorder the factors of te table:

You should begin with the sociodemographics: (age, gender (use sexe please), education,..children, ...). Then you return to: health status, COVID-19 infections, COVID-19 vaccination...).

Author comments: Thank you. We have updated it now.

  1. line 211: You should fine what do you consider as moderate knowledge, poor, adequate, (in the conclusion you used sufficient, appropriate.....). Define these terms. what is the rate that allowed you to define the level as adequate, poor....?. Later how did you define the factors associated with the high level of knowledge?. In the same line: what does this expression mean ? "knowledge (50-75% knowledge scores = 211 68.1%)".

Author comments: Thank you. We have mentioned this in results section. We have also updated the Methodology section to include these details as well, and hope this is now OK.

  1. Figure 1, 2 and 3: delete the title frm the backgound of the figures.

Author comments: Thank you. We have updated it now.

  1. Figure 1 and 2: reduce the size of the figures (you can make them in one line).

Author comments: Thank you. We have updated it now.

  1. Figure 3: delete the vertical bars .improve is quality by making each item in 1 line only (exp the four last items). complete the item and delete the points.

Author comments: Thank you. We have updated it now.

  1. Line 246: corect please
  2. Line 247: towards vaccination.
  3. Lines 248-152: the sentences are incomplete
  4. Line 256: practice preventive measures is not correct
  5. Line 260-264: this may be also to the fact that the study was conducted in 2022 (not in 2020), effect od scial medias, medias....
  6. Line 270: delete "is"
  7. Line 271: delete the sentence "we are....studies".
  8. Line 273: delete "is'", correct "control". also you should delete al the paragraph of lines 273-76. (no relation)
  9. Lines 278-281: there interminable number of studies about vaccine hesitancy....
  10. Line 282: 83: correct the numbers (25.06....)
  11. Line 282-294: multiple redundant and contradictory sentences. Revise
  12. Line 307-312: delete the paragraph

Author comments: Thank you for these comments. As seen – we have now completely re-organised the Discussion section and hope this is now acceptable.

  1. Line 331:  "only a quarter of children in Pakistan" i may be a quarter of the study population but not all the children of Pakistan.

Author comments: Thank you. We have revised it now.

  1. In conclusion: The manuscript requires an extensive revision

Author comments: Thank you – now extensively revised. We trust this is now OK.

Reviewer 3 Report

The manuscript entitled "Awareness and Practices towards vaccinating their children against COVID-19: A Cross-Sectional study among parents by Harmain Z. U at al analyzes the attitude of parents in the Punjab province of Pakistan towards vaccinating their children (aged 1 to 18 years) against COVID-19.

The study design is appropriate and rigorous.  The study findings are presented very clearly. 

Major Objections:

1.1  The authors take the stance  and conclude that vaccine hesitancy towards minors is due to "misinformation" and "false beliefs" and do not entertain the fact that these parents generally made a good decision for themselves and were largely vaccinated and could therefore have been equally wise in refraining from vaccinating their children.   I do not agree with their conclusion.  Most European countries suspended vaccination of minors over a year ago and most have stopped recommending these vaccinations except for those most at risk.  Moreover, various adverse effects of the vaccines continue to emerge, which has resulted in the Astra-Zeneca and Johnson and Johnson vaccinnes being pulled from various western markets.  The mRNA vaccines / gene therapy are a new class of vaccine that previously had not met the requirements for clinical trial approval and were able to do so for COVID-19 under emergency-use authorization, so clearly the COVID-19 situation was exceptional in terms of how quickly the vaccines went through trial and how quickly a new class of vaccines was approved. 

Now that the pandemic is officially over, these emergency-use recommendations need to be revised.  I believe these parents had very legitimate concerns that the authors fail to acknowledge and discuss.  What is the Pakistani government policy or vaccinating Children for COVID-19, in the light of these recent developments and why does their stance differ from European countries in general?  A key manuscript that the authors could refer to is Loannidis et al. 2020 - https://www.sciencedirect.com/science/article/pii/S0013935120307854?via%3Dihub, that indicates that risk of death after COVID infection is low in people aged 65 and under with no underlying co-morbidities.  Moreover, the originally reported efficacy of the vaccines has waned considerably as the virus has evolved.   It seems to me like the parents in this study could have been making and are continuing to make a reasonable decision not to vaccinate their children considering the risks of adverse effects versus benefit of the vaccine. 

Minor issues that need to be addressed include: - 

1.2. It would be good to have access to the full interview via a supplementary table, perhaps highlighting the modifications that were made compared to the original questionnaire that it was derived from.

1.3. Generally the results were clearly presented.  The only minor critiques are that Figures 1 and 2 take up too much space and could be combined into one composite figure and reduced in size.  Figure 3 could also be reduced in size.

Thank You.

2.1 Overall the text could do with proof-reading and editing from a professional science writing editor to eliminate various grammatical errors and improve readability.  I will give examples of this and other minor errors that should be addressed below too.

For example: - 

*Lines 33 - 34:  The sentence beginning with Consequently is incomplete.  It should state what the authors wanted to get a better understanding of e.g., 'Consequently, the objective of this study was to gain a better understanding of the factors preventing parents from vaccinating their Children in Pakistan.'

*Line 36 should read:  43.2% belonged to the 40-49 age group...

*Line 51:  It should be noted that SARS-CoV-2 is the name of the virus, whereas COVID-19 is the name of the disease. This sentence would read better as something like 'Since the emergence of the SARS-CoV-2 virus and the associated disease COVID-19, in Wuhan, China, in 2019....'

*Line 62 to 65:  These two sentences seem to contradict each other.  The first sentence claims that mortality rates are lower in hospitalized children in LMICs while the second sentence claims the opposite.

*Line 74:  Should read ...'Punjab Province', not, 'Provice Punjab'...

*Lines 76 to 77: Would read better as '... among neonates and children admitted WITH COVID-19 in different referral hospitals, 3.2% had died[16].'

*Line 88: should read 'rigorous campaign', not vigorous...

and the list goes on.

2.2  The abstract could be organized and edited to read more clearly.  The results sections generally flows well, so authors should follow the same style for the Abstract.  I personally would like to see the actual numbers reported for each category with percentages in bracket, for full transparency.  Authors should stick to the same style for reporting figures through out rather than giving percentages in brackets in some instances, then stating that "overall, only a quarter... in  line 39.

Author Response

Reviewer 3

Open Review

Quality of English Language

( ) I am not qualified to assess the quality of English in this paper
( ) English very difficult to understand/incomprehensible
( ) Extensive editing of English language required
(x) Moderate editing of English language required
( ) Minor editing of English language required
( ) English language fine. No issues detected

Author comments: Thank you for this. The manuscript has been edited by one of the co-authors who is a native English speaker with over 500 publications in recent years in peer-reviewed Journals. We trust this is now acceptable.

Yes

Can be improved

Must be improved

Not applicable

Does the introduction provide sufficient background and include all relevant references?

(x)

( )

( )

( )

Are all the cited references relevant to the research?

(x)

( )

( )

( )

Is the research design appropriate?

(x)

( )

( )

( )

Are the methods adequately described?

(x)

( )

( )

( )

Are the results clearly presented?

( )

(x)

( )

( )

Are the conclusions supported by the results?

( )

(x)

( )

( )

Comments and Suggestions for Authors

The manuscript entitled "Awareness and Practices towards vaccinating their children against COVID-19: A Cross-Sectional study among parents by Harmain Z. U at al analyses the attitude of parents in the Punjab province of Pakistan towards vaccinating their children (aged 1 to 18 years) against COVID-19.The study design is appropriate and rigorous.  The study findings are presented very clearly. 

Author comments: Thank you for your kind comments – appreciated. We also thank you for your suggestions that have enhanced the paper. We hope we have now adequately addressed your queries.

Major Objections:

1.1 The authors take the stance and conclude that vaccine hesitancy towards minors is due to "misinformation" and "false beliefs" and do not entertain the fact that these parents generally made a good decision for themselves and were largely vaccinated and could therefore have been equally wise in refraining from vaccinating their children. I do not agree with their conclusion.  Most European countries suspended vaccination of minors over a year ago and most have stopped recommending these vaccinations except for those most at risk.  Moreover, various adverse effects of the vaccines continue to emerge, which has resulted in the Astra-Zeneca and Johnson and Johnson vaccines being pulled from various western markets.  The mRNA vaccines / gene therapy is a new class of vaccine that previously had not met the requirements for clinical trial approval and were able to do so for COVID-19 under emergency-use authorization, so clearly the COVID-19 situation was exceptional in terms of how quickly the vaccines went through trial and how quickly a new class of vaccines was approved. 

Now that the pandemic is officially over, these emergency-use recommendations need to be revised.  I believe these parents had very legitimate concerns that the authors fail to acknowledge and discuss.  What is the Pakistani government policy or vaccinating Children for COVID-19, in the light of these recent developments and why does their stance differ from European countries in general?  A key manuscript that the authors could refer to is Loannidis et al. 2020 - https://www.sciencedirect.com/science/article/pii/S0013935120307854?via%3Dihub, that indicates that risk of death after COVID infection is low in people aged 65 and under with no underlying co-morbidities.  Moreover, the originally reported efficacy of the vaccines has waned considerably as the virus has evolved.   It seems to me like the parents in this study could have been making and are continuing to make a reasonable decision not to vaccinate their children considering the risks of adverse effects versus benefit of the vaccine. 

Author comments: Thank you for the comment. Overall, vaccines hesitancy is quite common in Pakistan due to many reasons including a lack of safety data, religious misbeliefs, improper awareness among mass besides misinformation. However studies undertaken in Pakistan with respect to the vaccine have shown only a limited number of vaccinated individuals have experienced any side effects from the vaccine, and most of the side effects seen are minor with patients not needing medicines and/ or hospitalization for their management. As far as vaccines policy an recommendation are concerned, the Government of Pakistan recommends that every child aged above five years should be vaccinated as part of the ongoing COVID-19 vaccination campaigns (now with an additional reference). Thank you for the reference of Loannidis et al. 2020 which we have incorporated in the Discussion. However – also discussed in the Discussion are the current mortality rates in children (up to March 2023) as well as the fact that vaccinating children also impacts on others, etc. We hope this is now acceptable.

Minor issues that need to be addressed include: - 

1.2. It would be good to have access to the full interview via a supplementary table, perhaps highlighting the modifications that were made compared to the original questionnaire that it was derived from.

Author comments: Thank you. We have provided study questionnaire as supplementary file along with others details.

1.3. Generally the results were clearly presented.  The only minor critiques are that Figures 1 and 2 take up too much space and could be combined into one composite figure and reduced in size.  Figure 3 could also be reduced in size.

Author comments:  Thank You. We have updated figures 1, 2, and 3.

Comments on the Quality of English Language

2.1 Overall the text could do with proof-reading and editing from a professional science writing editor to eliminate various grammatical errors and improve readability.  I will give examples of this and other minor errors that should be addressed below too. For example: - 

Author comments: Thank you for this. We have now gone through the paper with the help of one of the co-authors who is a native English speaker with over 500 publications in peer-reviewed Journals to his name. We trust this is now OK.

*Lines 33 - 34:  The sentence beginning with Consequently is incomplete.  It should state what the authors wanted to get a better understanding of e.g., 'Consequently, the objective of this study was to gain a better understanding of the factors preventing parents from vaccinating their Children in Pakistan.'

Author comment. Thank you. We have updated it now

*Line 36 should read:  43.2% belonged to the 40-49 age group...

Author comment. Thank you. We have updated it now

*Line 51:  It should be noted that SARS-CoV-2 is the name of the virus, whereas COVID-19 is the name of the disease. This sentence would read better as something like 'Since the emergence of the SARS-CoV-2 virus and the associated disease COVID-19, in Wuhan, China, in 2019....'

Author comment. Thank you. We have updated it now

*Line 62 to 65:  These two sentences seem to contradict each other.  The first sentence claims that mortality rates are lower in hospitalized children in LMICs while the second sentence claims the opposite.

Author comment. Thank you – now refined as missed off ICUs. We hope this is now OK.

*Line 74:  Should read ...'Punjab Province', not, 'Provice Punjab'...

Author comment. Thank you. We have updated it now

*Lines 76 to 77: Would read better as '... among neonates and children admitted WITH COVID-19 in different referral hospitals, 3.2% had died[16].'

Author comment. Thank you. We have updated it now

*Line 88: should read 'rigorous campaign', not vigorous...

and the list goes on.

Author comment. Thank you. We have updated it now

2.2  The abstract could be organized and edited to read more clearly.  The results sections generally flows well, so authors should follow the same style for the Abstract.  I personally would like to see the actual numbers reported for each category with percentages in bracket, for full transparency.  Authors should stick to the same style for reporting figures through out rather than giving percentages in brackets in some instances, then stating that "overall, only a quarter... in  line 39.

Author comment. Thank you. We have updated it, and hope this is now acceptable.

Round 2

Reviewer 2 Report

I would thank the authors for the efforts made to improve the quality of the manuscript. It is really improved regarding the language and the way of writing. However, I still have some concerns:

1. The first concern is related to the statistical analysis. Are you sure that thés tests (MW and KW) are adapted for your results? These tests are used for scores not for proportions. Be careful and confirm. 

2. You should add subtiles to separate your results.  Also, avoid to use to use to figures and/or tables consecutively. Furthermore, you can make figures 1 and 2 in one line.

In figure 3, you can class the items according to their percentages.

I have also some minor remarks:

- line 40 : add that after considered.

- line 137: add the subject of the sentence after Subsequently

- delete the sentence of lines 187-191.

Minor 

Author Response

Reviewer 2

Open Review

Quality of English Language

( ) I am not qualified to assess the quality of English in this paper
( ) English very difficult to understand/incomprehensible
( ) Extensive editing of English language required
( ) Moderate editing of English language required
(x) Minor editing of English language required
( ) English language fine. No issues detected

Author comments: Thank you for this. We have now been through the manuscript with the help of one of the co-authors who is a native English speaker with over 500 publications in peer-reviewed Journals. We trust this is now acceptable.

Yes

Can be improved

Must be improved

Not applicable

Does the introduction provide sufficient background and include all relevant references?

(x)

( )

( )

( )

Are all the cited references relevant to the research?

(x)

( )

( )

( )

Is the research design appropriate?

( )

(x)

( )

( )

Are the methods adequately described?

( )

(x)

( )

( )

Are the results clearly presented?

( )

(x)

( )

( )

Are the conclusions supported by the results?

(x)

( )

( )

( )

Comments and Suggestions for Authors

I would thank the authors for the efforts made to improve the quality of the manuscript. It is really improved regarding the language and the way of writing.

Author comments: Thank you for these kind words – appreciated!

However, I still have some concerns:

  1. The first concern is related to the statistical analysis. Are you sure that thés tests (MW and KW) are adapted for your results? These tests are used for scores not for proportions. Be careful and confirm. 

Author comments: Respected Reviewer, we used Mann-Whitney U test to compare knowledge and practices scores between the vaccine acceptors and rejecters (see Table 5). We have clarified this in the statistical analysis.  We trust this is now OK.

  1. You should add subtiles to separate your results.  Also, avoid to use to use to figures and/or tables consecutively. Furthermore, you can make figures 1 and 2 in one line. In figure 3, you can class the items according to their percentages.

Author comments: Thank you for these comments. Figure 2 and 3 have now been revised per your suggestion. Regarding the consecutive placement of figures and Tables, the MDPI production office usually settles this by placing Tables and figures at appropriate places in the manuscript. We hope this is now acceptable.

3) I have also some minor remarks:

- line 40 : add that after considered.

- line 137: add the subject of the sentence after Subsequently

Author comments: Thank you now changed.

- delete the sentence of lines 187-191.

Author comments: Thank you for this. We would like to keep the comment regarding the Knowledge scores if we can so people are aware of our methodology when reading the findings. We trust this is OK. We would also like to keep the last sentence if we can – again to inform readers regarding the basis of our recommendations (building on comments from other Reviewers). We trust this is also acceptable.

Comments on the Quality of English Language - Minor 

Author comments: Thank you for this. We have now been through the manuscript with the help of one of the co-authors who is a native English speaker with over 500 publications in peer-reviewed Journals. We trust this is now acceptable.